# An International Comparison of Presentation, Outcomes and CORONET Predictive Score Performance in Patients with Cancer Presenting with COVID-19 across Different Pandemic Waves

**DOI:** 10.3390/cancers14163931

**Published:** 2022-08-16

**Authors:** Oskar Wysocki, Cong Zhou, Jacobo Rogado, Prerana Huddar, Rohan Shotton, Ann Tivey, Laurence Albiges, Angelos Angelakas, Dirk Arnold, Theingi Aung, Kathryn Banfill, Mark Baxter, Fabrice Barlesi, Arnaud Bayle, Benjamin Besse, Talvinder Bhogal, Hayley Boyce, Fiona Britton, Antonio Calles, Luis Castelo-Branco, Ellen Copson, Adina Croitoru, Sourbha S. Dani, Elena Dickens, Leonie Eastlake, Paul Fitzpatrick, Stephanie Foulon, Henrik Frederiksen, Sarju Ganatra, Spyridon Gennatas, Andreas Glenthøj, Fabio Gomes, Donna M. Graham, Christina Hague, Kevin Harrington, Michelle Harrison, Laura Horsley, Richard Hoskins, Zoe Hudson, Lasse H. Jakobsen, Nalinie Joharatnam-Hogan, Sam Khan, Umair T. Khan, Khurum Khan, Alexandra Lewis, Christophe Massard, Alec Maynard, Hayley McKenzie, Olivier Michielin, Anne C. Mosenthal, Berta Obispo, Carlo Palmieri, Rushin Patel, George Pentheroudakis, Solange Peters, Kimberly Rieger-Christ, Timothy Robinson, Emanuela Romano, Michael Rowe, Marina Sekacheva, Roseleen Sheehan, Alexander Stockdale, Anne Thomas, Lance Turtle, David Viñal, Jamie Weaver, Sophie Williams, Caroline Wilson, Caroline Dive, Donal Landers, Timothy Cooksley, André Freitas, Anne C. Armstrong, Rebecca J. Lee

**Affiliations:** 1Department of Computer Science, University of Manchester, Oxford Road, Manchester M13 9PL, UK; oskar.wysocki@manchester.ac.uk (O.W.); andre.freitas@manchester.ac.uk (A.F.); 2Digital Experimental Cancer Medicine Team, Cancer Biomarker Centre, Cancer Research UK Manchester Institute, University of Manchester, Alderley Park, Macclesfield SK10 4TG, UK; paul.fitzpatrick@digitalecmt.org (P.F.); 3Cancer Research UK Manchester Institute, Cancer Biomarker Centre, The University of Manchester, Al-derley Park, Macclesfield SK10 4TG, UK; cong.zhou@cruk.manchester.ac.uk (C.Z.); caroline.dive@manchester.ac.uk (C.D.); 4Department of Medical Oncology, Hospital Universitario Infanta Leonor, Av. Gran Vía del Este, 80, 28031 Madrid, Spain; jacobo.rogado@gmail.com (J.R.); berta.obispo@gmail.com (B.O.); 5Department of Medical Oncology, The Christie NHS Foundation Trust, Wilmslow Road, Manchester M20 4BX, UK; prerana.huddar@nhs.net (P.H.); rshotton@nhs.net (R.S.); ann.tivey@nhs.net (A.T.); ange-los.angelakas@nhs.net (A.A.); kathryn.banfill@manchester.ac.uk (K.B.); fiona.britton1@nhs.net (F.B.); fabio.gomes2@nhs.net (F.G.); donna.graham8@nhs.net (D.M.G.); christi-na.hague@christie.nhs.uk (C.H.); laura.horsley@nhs.net (L.H.); alexandra.lewis12@nhs.net (A.L.); ja-mie.weaver2@nhs.net (J.W.); tim.cooksley@nhs.net (T.C.); anne.armstrong3@nhs.net (A.C.A.); 6Faculty of Biology, Medicine and Health, The University of Manchester, Oxford Road, Manchester M13 9PL, UK; oskar.wysocki@manchester.ac.uk (O.W.);; 7Department of Medical Oncology, Gustave Roussy, 94805 Villejuif, France; laurence.albiges@gustaveroussy.fr (L.A.); fabrice.barlesi@gustaveroussy.fr (F.B.); 8Department of Oncology, Haematology and Palliative Care, Asklepios Klinik Altona, Paul-Ehrlich-Str. 1, 22763 Hamburg, Germany; dirk.arnold@arcor.de; 9Weston Park Cancer Centre, Sheffield Teaching Hospitals NHS Foundation Trust, Sheffield S10 2JF, UK; aungtg@gmail.com (T.A.); hayley.boyce@nhs.net (H.B.); alec.maynard@nhs.net (A.M.); roseleen.sheehan@nhs.net (R.S.); sophie.williams42@nhs.net (S.W.); caroline.wilson42@nhs.net (C.W.); 10Division of Molecular and Clinical Medicine, Ninewells School of Medicine, University of Dundee, Dundee DD2 1SY, UK; m.z.baxter@dundee.ac.uk; 11Multidisciplinary Oncology & Therapeutic Innovations Department, Aix Marseille University, CNRS, INSERM, CRCM, 13015 Marseille, France; 12Drug Development Department (DITEP), Gustave Roussy—Cancer Campus, 94805 Villejuif, France; arnaud.bayle@gustaveroussy.fr (A.B.); benjamin.besse@gustaveroussy.fr (B.B.); christophe.massard@gustaveroussy.fr (C.M.); 13Oncostat (CESP U1018 INSERM), Labeled Ligue Contre le Cancer, University Paris-Saclay, 94805 Villejuif, France; stephanie.foulon@gustaveroussy.fr; 14Department of Medical Oncology, The Clatterbridge Cancer Centre NHS Foundation Trust, 65 Pembroke Place, Liverpool L7 8YA, UK; talvinder.bhogal@nhs.net (T.B.); umairkhan@nhs.net (U.T.K.); c.palmieri@liverpool.ac.uk (C.P.); 15Department of Medical Oncology, Hospital General Universitario Gregorio Marañón, Calle del Dr. Esquerdo, 46, 28007 Madrid, Spain; antonio.calles@live.com; 16ESMO-CoCARE Steering Committee, European Society for Medical Oncology, Via Ginevra 4, 6900 Lugano, Switzerland; luismocb@hotmail.com (L.C.-B.); george.pentheroudakis@esmo.org (G.P.); solange.peters@chuv.ch (S.P.); 17NOVA National School of Public Health, Av. Padre Cruz, 1600-560 Lisboa, Portugal; 18Department of Medical Oncology, University Hospital Center of Algarve, R. Leao Penedo, 8000-386 Faro, Portugal; 19Cancer Sciences Academic Unit, University Hospital Southampton NHS Foundation Trust, Tremona Road, Southampton SO16 6YD, UK; e.copson@soton.ac.uk (E.C.); hayley.mckenzie@uhs.nhs.uk (H.M.); 20Medical Oncology Department, Fundeni Clinical Institute, 258, Fundeni Str., 022238 București, Romania; adina.croitoru09@yahoo.com; 21Department of Cardiology, Lahey Hospital and Medical Center, Burlington, MA 01805, USA; sourbha.s.dani@lahey.org (S.S.D.); sarju.ganatra@lahey.org (S.G.); anne.c.mosenthal@lahey.org (A.C.M.); rushinwildlife@gmail.com (R.P.); kimberly.r.christ@lahey.org (K.R.-C.); 22Oncology Department, University Hospitals of Leicester NHS Trust, Leicester LE1 5WW, UK; elenadickens89@gmail.com (E.D.); sk504@leicester.ac.uk (S.K.); at107@leicester.ac.uk (A.T.); 23Department of Medical Oncology, University Hospitals Plymouth NHS Trust, Derriford Road, Plymouth PL6 8DH, Devon, UK; leonie.eastlake@nhs.net; 24Biostatistics and Epidemiology Office, Gustave Roussy, University Paris-Saclay, 94805 Villejuif, France; 25Department of Haematology, Odense University Hospital, 5000 Odense, Denmark; henrik.frederiksen@rsyd.dk; 26Department of Medical Oncology, The Royal Marsden NHS Foundation Trust, Fulham Road, London SW3 6JJ, UK; spyridon.gennatas@rmh.nhs.uk (S.G.); kevin.harrington@icr.ac.uk (K.H.); 27Department of Haematology, Copenhagen University Hospital—Rigshospitalet, 2100 Copenhagen, Denmark; andreas.glenthoej@regionh.dk; 28The Institute of Cancer Research NIHR Biomedical Research Centre, London SW3 6JB, UK; 29Ninewells Hospital and Medical School, Dundee DD2 1SG, UK; michelle.harrison2@nhs.scot; 30Research IT, University of Manchester, Oxford Road, Manchester M13 9PL, UK; richard.p.hoskins@manchester.ac.uk; 31Bristol Haematology and Oncology Centre, University Hospitals Bristol NHS Foundation Trust, Bristol BS2 8ED, UK; zoe.hudson@uhbw.nhs.uk (Z.H.); tim.robinson@uhbw.nhs.uk (T.R.); 32Department of Haematology, Clinical Cancer Research Center, Aalborg University Hospital, 9000 Aalborg, Denmark; lasse.j@rn.dk; 33Department of Medical Oncology, University College London Hospitals NHS Foundation Trust, 235 Euston Road, London NW1 2BU, UK; n.joharatnam@nhs.net (N.J.-H.); khurum.khan1@nhs.net (K.K.); 34MRC Clinical Trials Unit, University College London, Gower St., London WC1E 6BT, UK; 35Institute of Systems, Molecular and Integrative Biology, The University of Liverpool, Liverpool L69 3BX, UK; 36Department of Oncology, Melanoma Clinic, Swiss Institute of Bioinformatics, Quartier Sorge—Batiment Amphipole, 1015 Lausanne, Switzerland; olivier.michielin@chuv.ch; 37Medical Oncology, Centre Hospitalier Universitaire Vaudois, Rue du Bugnon 46, 1011 Lausanne, Switzerland; 38Population Health Sciences, University of Bristol, Oakfield House, Oakfield Grove, Bristol BS8 2BN, UK; 39Department of Oncology, Institut Curie, PSL Research University, Office 2A-5, 26, Rue d’Ulm, 75005 Paris, France; emanuela.romano@curie.fr; 40Sunrise Centre, Royal Cornwall Hospitals NHS Trust, Truro TR1 3LJ, Cornwall, UK; michael.rowe2@nhs.net; 41World-Class Research Center “Digital Biodesign and Personalized Healthcare”, Sechenov First Moscow State Medical University, 119991 Moscow, Russia; sekach_rab@mail.ru; 42Tropical and Infectious Diseases Unit, Royal Liverpool University Hospital, 3Z Link, Prescot Street, Liverpool L7 8XP, UK; a.stockdale@liverpool.ac.uk (A.S.); lance.turtle@liverpool.ac.uk (L.T.); 43Leicester Cancer Research Centre, The University of Leicester, University Road, Leicester LE1 7RH, UK; 44Department of Medical Oncology, Hospital Universitario La Paz, Paseo de la Castellana, 261, 28046 Madrid, Spain; dvinallozano@gmail.com; 45Idiap Research Institute, 1920 Martigny, Switzerland

**Keywords:** COVID-19, cancer, CORONET, outcomes, Omicron, vaccination

## Abstract

**Simple Summary:**

There have been huge improvements in both vaccination and the management of COVID-19 in patients with cancer. In addition, different variants may be associated with different presentations. Therefore, we examined whether indicators of the severity of COVID-19 in patients with cancer who present to hospital varied during different waves of the pandemic and we showed that these indicators remained predictive. We validated that the COVID-19 Risk in Oncology Evaluation Tool (CORONET), which predicts the severity of COVID-19 in cancer patients presenting to hospital, performed well in all waves. In addition, we examined patient outcomes and the factors that influence them and found that there was increased vaccination uptake and steroid use for patients requiring oxygen in later waves, which may be associated with improvements in outcome.

**Abstract:**

Patients with cancer have been shown to have increased risk of COVID-19 severity. We previously built and validated the COVID-19 Risk in Oncology Evaluation Tool (CORONET) to predict the likely severity of COVID-19 in patients with active cancer who present to hospital. We assessed the differences in presentation and outcomes of patients with cancer and COVID-19, depending on the wave of the pandemic. We examined differences in features at presentation and outcomes in patients worldwide, depending on the waves of the pandemic: wave 1 D614G (n = 1430), wave 2 Alpha (n = 475), and wave 4 Omicron variant (n = 63, UK and Spain only). The performance of CORONET was evaluated on 258, 48, and 54 patients for each wave, respectively. We found that mortality rates were reduced in subsequent waves. The majority of patients were vaccinated in wave 4, and 94% were treated with steroids if they required oxygen. The stages of cancer and the median ages of patients significantly differed, but features associated with worse COVID-19 outcomes remained predictive and did not differ between waves. The CORONET tool performed well in all waves, with scores in an area under the curve (AUC) of >0.72. We concluded that patients with cancer who present to hospital with COVID-19 have similar features of severity, which remain discriminatory despite differences in variants and vaccination status. Survival improved following the first wave of the pandemic, which may be associated with vaccination and the increased steroid use in those patients requiring oxygen. The CORONET model demonstrated good performance, independent of the SARS-CoV-2 variants.

## 1. Introduction

The SARS-CoV-2 virus, which causes COVID-19 disease, has infected over half a billion people worldwide, resulting in at least 6 million deaths [1]. Patients with cancer have been particularly affected, in terms of both cancer treatment delays and risks of more severe COVID-19 disease. Early in the pandemic, studies reported mortality rates of between 10% and 30% [2,3,4,5,6,7]. Thereafter, vaccines have been rapidly developed and a number of disease-modifying treatments have been identified. Vaccinations have resulted in decreased numbers of patients experiencing severe COVID-19; however, vaccinations have been less effective in reducing infection rates, particularly with respect to the newer Omicron variants [8,9,10]. Upon presentation with COVID-19, depending on the variant and the severity of infection, patients may receive corticosteroids, antivirals such as remdesivir, or antibodies such as tocilizumab [11,12,13]. Together, these have improved outcomes, at least in the developed world [11,12,13].

Despite the rapid progress in the prevention and management of COVID-19, patients with cancer continue to present to hospital with severe COVID-19 disease. Studies have shown that the severity of COVID-19 is similar in patients who are infected with different SARS-CoV-2 variants; however, there are improved outcomes when patients with cancer are vaccinated, compared with patients who remain unvaccinated [14]. At the start of the pandemic, numerous studies examined the associations between different clinical features and the severity of COVID-19 [2,4,6,7,15,16]. However, it is not clear whether these associations remain discriminatory in patients who are infected with different SARS-CoV-2 variants and in patients who are vaccinated. Therefore, we examined whether clinical features at presentation to hospital changed over time, based on the increased uptake of vaccinations and on infection by different coronavirus variants. In addition, we examined whether patient outcomes changed following presentation to hospital with COVID-19 and cancer. Previously, we developed the COVID-19 Risk in Oncology Evaluation Tool (CORONET) to guide healthcare professionals and systems in decision-making regarding the need for admission and to provide information regarding the likely severity of COVID-19 illness [17]. This tool allowed us to examine whether there were differences in the severity of infection, determined by CORONET scores during different waves of the pandemic. Having previously shown that the CORONET tool performed well for patients with D614G SARS-CoV-2 and Alpha variants, we validated its usage for patients with Omicron infections.

## 2. Materials and Methods

Approval (reference 20/WA/0269) was granted from the UK Research Ethics Committee for this study. Information regarding governance/regulatory approvals for each international cohort are available in previously published papers [17]. Patients with active cancer were included in the study, We defined active cancer as solid (stage I–IV) or haematological cancer diagnosed in the last 6 months or cancer or recurrent or metastatic or haematological cancer, not in complete remission, that was treated for 6 months or more. The selected patients had a laboratory-confirmed SARS-CoV-2 infection. They were defined as vaccinated if they had more than one dose of any available vaccination against SARS-CoV-2. No patients in the dataset were infected more than one time; thus, each patient presented with a unique case of COVID-19. Asymptomatic patients who were screened as part of routine testing for surgical procedures and found to be positive were not included, as their data were not routinely captured. Details regarding data collection and outcome definitions are provided in Lee et al. [17].

Patient data were collected worldwide from the United Kingdom, the United States, Spain, Denmark, and France, and from medical centres contributing to ESMO-CoCARE. We included a range of hospitals that manage cancer patients in different settings, from local district general/local community hospitals, in which general acute physicians manage acute oncology admissions, to tertiary cancer centres, within which there are highly specialised oncology services [17].

We defined the different dominant variants responsible for the SARS-CoV-2 waves as follows: the wave 1 D614G SARS-CoV-2 variant, including patients who presented from March 2020 until the end of August 2020; the wave 2 Alpha variant, comprising patients who presented from September 2020 until the end of April 2021; and wave 4 Omicron variant from December 2021 until the end of February 2022 (the date of cut-off for data collection) [1]. Of note, we did not include the Delta wave 3, due to the insufficiency of available data (n = 15) to support meaningful conclusions.

We previously developed CORONET to help determine the need to admit patients to hospital on the bases of their likelihood of needing oxygen (as generally oxygen is only given in hospital) and their severity of COVID-19, as indicated by predictions for required oxygen and/or death [17]. As a result, we established four key outcomes, arranged in a 0–3 point ordinal scale: discharged, admitted (≥24 h inpatient), admitted+O_2_ requirement (including ventilator support), and admitted+O_2_+died (with the death directly attributable to COVID-19 disease, not to cancer). These four outcomes were used as measures of disease severity [17]. Contrary to an analysis of binary outcomes (e.g., need for oxygen vs. no need for oxygen), this strategy improved the ability to provide a complete clinical picture that was important for overall decision-making regarding hospital admission. A threshold of ≥1.0 determined admissions and a threshold of ≥2.3 determined severe outcome (additional details are available in [17]). 

Differences between waves and in outcomes were assessed using the Pearson’s Chi-square (Χ^2^) test and the Mann–Whitney U test. We evaluated the differences in both the CORONET scores and in the performances of the model during different waves. Patients who missed more than one variable from the five features that provided the greatest contribution to the score (i.e., National Early Warning Score 2 (NEWS2), C-reactive protein (CRP), albumin, age, and platelets) were excluded from the CORONET score evaluation. The significance level was set at *p* = 0.05, and *p* values were adjusted for multiple testing using the Benjamini–Hochberg method. CORONET’s performance on clinical decision-making for patients presenting with different SARS-CoV-2 variants was examined using the receiver operating characteristic (ROC). CORONET scores were calculated for patients that were not in the model’s derivation cohort (see [17]). 

## 3. Results

Patient data were collected for wave 1 D614G (n = 1430) and wave 2 Alpha (n = 475) from 12 participating hospitals in the UK, two hospitals in Spain, four hospitals in the USA, and (via the ESMO-CoCARE registry) hospitals throughout the world, excluding the USA, Canada, and Latin America (see [17]). Patients in wave 4 Omicron (n = 63) were treated in the UK and Spain (details regarding number of patients are provided in Appendix A). Clinical features of the patients are shown in Table 1. Intriguingly, patients who presented to hospital in wave 2 had a significantly lower median age, compared with first wave patients (61 vs. 68 *p* < 0.001, Table 1). Almost all patients (i.e., 95.8%) in the UK/Spain cohort who presented to hospital with available data (48 of the 63, with 15 having missing data) were vaccinated with at least one dose by the time the Omicron wave developed.

We observed that there were significantly more patients discharged within 24 h in waves 2 and 4, compared with patients discharged within 24 h in wave 1 (186, or 39.2%, 26, or 41.3%, and 336. or 23.5%, respectively; see Table 2). The mortality rate for patients admitted with COVID-19 and cancer was 26.4% in wave 1 vs. 13.1% in wave 2 and 0% in wave 4 (wave 2 vs. wave 1; 1 *p* < 0.001, with no deaths observed in the UK/Spain Omicron cohort). We observed no significant difference in the requirements for oxygen between the waves (wave 1 vs. wave 2, *p* = 0.265; wave 1 vs. wave 4, *p* = 0.368; wave 2 vs. wave 4, *p* = 0.265). Steroids were the mainstay of treatment following hospital admission, particularly in waves 2 and 4. Antivirals and antibodies were rarely used (Table 3). Of note, 40% (32/81) of patients requiring oxygen in wave 1, 88% (78/89) of patients requiring oxygen in wave 2, and 94% (15/16) of patients requiring oxygen in wave 4 were treated with steroids, showing the impact of the RECOVERY trial data (the use of steroids only for those patients requiring oxygen) in the management of COVID-19 [12].

Next, we examined whether there were differences in patients’ clinical features during different waves of the pandemic (Table 4). In particular, patients with COVID-19 were stratified by treatment outcomes to explore differences that were specific to COVID-19 severity. Patients in wave 1 were older than patients in wave 2, independent of COVID-19 severity. Similarly, wave 1 patients presented with significantly lower cancer stages than patients in waves 2 or 4, independent of COVID-19 severity. Performance status differed only among patients who died due to COVID-19 in waves 1 and 2 (higher in wave 1, *p* = 0.0075). There were no significant differences during the three waves in the laboratory values of patients who presented with severe COVID-19 (patients who required oxygen or who were dying), apart from albumin, which was observed to be lower in patients requiring oxygen in wave 2 than in patients requiring oxygen in wave 1 (*p* = 0.0029).

We investigated whether the features remained significant in predicting the severity of COVID-19 during each wave (Figure 1, Appendix A). Older patients were significantly more likely to have severe outcomes in both waves 1 and wave 2, but there was no such correlation in wave 4. Similarly, lower lymphocytes and platelets were associated with increased severity in waves 1 and 2, but these associations were not observed in wave 4 (Figure 1; for *p*-values see Appendix A). Patients with higher NEWS2, CRP, and neutrophils, and lower albumin, had more severe outcomes in all waves.

We previously developed the CORONET score to predict the severity of COVID-19 upon patients’ presentation to hospital and to aid in deciding whether to admit or discharge patients with cancer and COVID-19 [17]. In this analysis, CORONET scores were calculated for 258, 48, and 54 patients who were included from waves 1, 2, and 4, respectively, following exclusions due to missing values (Appendix A). Intriguingly, the CORONET scores during the three waves were not significantly different (*p* > 0.05, Appendix A, Figure 2), indicating consistent patient severity at the times of presentation and admission to hospital for all variants (Appendix A). Treatments for COVID-19 for each cohort are provided in Appendix A; steroid use was the treatment that differed most significantly. With consistent area under the curve (AUC) during the different waves, CORONET demonstrated its continuing capability to predict COVID-19 severity (Figure 3). The AUC for admission was 0.82 for wave 1 vs. 0.72 for wave 2 vs. 0.80 for wave 4; the AUC for death was 0.77 for wave 1 vs. 0.78 for wave 2 vs. 0.88 for wave 4. Sensitivity, specificity, positive predictive value, and negative predictive value for the CORONET scores in the different waves, based on the cutoffs determined in [17], are presented in Appendix A. CORONET scores indicated that admission was recommended for 94% of patients who required oxygen or died in wave 1, for 89% of patients who required oxygen or died in wave 2 and for 89% of patients who required oxygen or died in wave 4 (Appendix A and Figure 2). 

## 4. Discussion

Multiple studies have shown that patients with cancer have more severe outcomes due to COVID-19 than do patients without cancer, due to a combination of immune dysregulation and the effects of certain treatments, particularly those that deplete B cells in haematological cancers [18,19,20]. Over time, mortality in the general population versus mortality in members of the population who are infected has decreased during the different waves of the pandemic [21], which is likely due to multiple factors, including country-dependent public health measures, vaccination, and improved management of severe acute COVID-19 disease. However, patients with cancer, in general, have been shown to have reduced immune responses to vaccination and can present with severe COVID-19 [22,23,24]. It is important to note that following the first wave of the pandemic, the volume of cancer-directed therapies offered to patients returned to normal levels. Despite this, the mortality rate in cancer patients also decreased in the different waves, which is reassuring for patients who are undergoing cancer treatment.

Our study showed that features that were previously associated with COVID-19 severity in patients with cancer [2,6,25,26], such as low albumin, higher CRP, and neutrophils, remain discriminatory in patients who present with different variants. This is important in terms of assessing the likely severity of SARS-CoV-2 infections for individual patients. Indeed, we validated the CORONET score’s ability to identify patients with severe COVID-19 due to different variants and in patients with different vaccination status. The AUC was 0.80 for admission and 0.87 for the requirement of oxygen in patients presenting with the Omicron variant. Given the rapidly changing nature of the COVID-19 pandemic, it is particularly important to assess the temporal stability of risk prediction models. Here, we showed that in a small validation cohort, CORONET has a similar performance during the different pandemic waves and in different countries.

Our study revealed that at time of presentation to hospital, markers of COVID-19 severity were similar during the different waves, despite the fact that the majority of the patients were vaccinated during the Omicron wave. This is intriguing, as it suggests that management of patients with COVID-19 and cancer, following their presentation with COVID-19 to hospital, might affect outcomes. Vaccination may also affect the likelihood of mortality in severe COVID-19 patients [14]; however, we note that there was an increased use of steroids (40% in wave 1 88% in wave 2, and 94% in wave 4) in patients requiring oxygen, which may have had a significant impact, as other treatments were rarely used in our cohorts. The improvement in mortality (26.5% in wave 1, 13.1% in wave 2), particularly in wave 2 where no patients were vaccinated, could be associated with steroid treatment. Therefore, studies evaluating the differences in COVID-19 outcomes over time should consider not only vaccination, but also post-infection management. However, there were significant amounts of missing data in our analysis, and to test these findings further, we propose to examine steroid use and mortality in vaccinated and non-vaccinated cancer patients, in larger cohorts. Based on the RECOVERY Phase III trial data in the general population and the interesting observations in our cohorts, we emphasise that it is essential that all patients with cancer and COVID-19, requiring oxygen, are managed with steroids unless contraindicated [12]. Of note, we observed that the disease stage of cancer in patients who present to hospital with COVID-19 was higher in waves 2 and 4 than in wave 1. This difference suggests that cancer severity may have an effect on the necessity for admission; however, the patients’ performance status was similar, which indicates that the situation is more complex. Larger cohorts are needed to consider this further. We also observed age differences in patients who presented during the different waves, with wave 2 patients having a lower median age. This may be due to the vaccine rollout, which targeted older patients first, or it may be that younger patients were more likely to become infected due to social factors, as observed in other studies [27].

The main limitation of this study was the small size of cohorts, particularly the cohort size of the Omicron wave. This factor may have affected the levels of significance for some of the statistical tests, particularly in the comparisons between wave 4 and wave 1. Significant data were missing, particularly in terms of the treatments used to manage COVID-19. In addition, due to this study’s retrospective nature, there was no centrally determined COVID-19 management plan; therefore, treatment of COVID-19 varied among the various treatment centres. Nonetheless, this study provides important provisional insights, associated with different pandemic waves, into the features of patients with cancer and COVID-19. These insights may be further examined in larger studies in the future.

## 5. Conclusions

Patients with cancer who present and are admitted to hospital with COVID-19 have similar features of severity, which remain discriminatory despite differences in SARS-CoV-2 variants and vaccination status. Survival improved following the first wave of the pandemic, which, at time of presentation to hospital, may be related to both vaccination and the increased steroid use in patients who require oxygen. The CORONET model demonstrated good performance, independent of the SARS-CoV-2 variants.

## Figures and Tables

**Figure 1 cancers-14-03931-f001:**
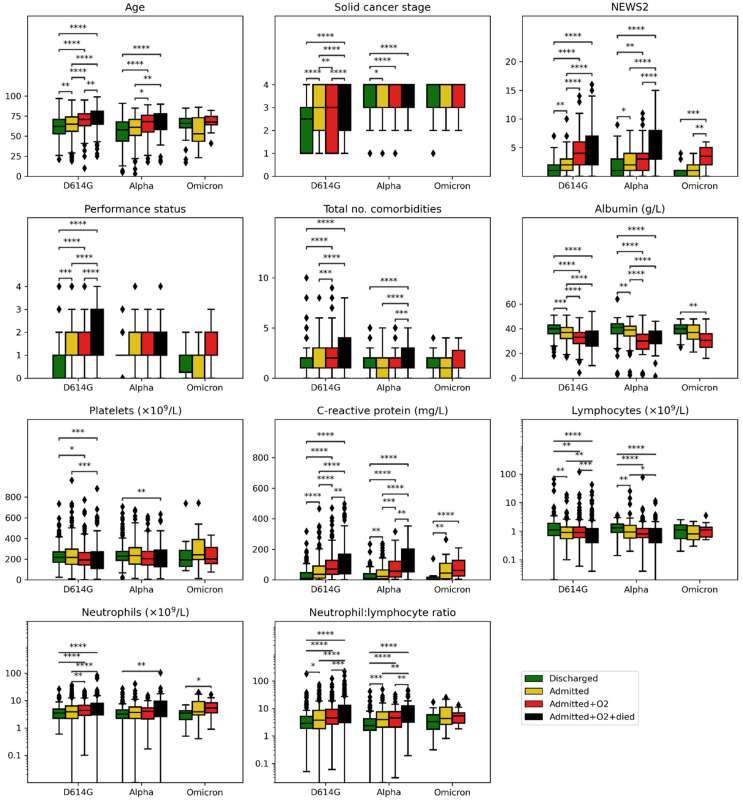
Known features associated with COVID-19 severity, compared during different waves and stratified by outcome. Statistically significant differences between features within the waves were marked by: ****—*p* < 0.0001, ***—*p* < 0.001, **—*p* < 0.01, *—*p* < 0.05; Mann–Whitney U and Chi^2^ tests were used for numeric and categorical (solid cancer stage and performance status) features, accordingly; *p* values were adjusted using the Benjamini–Hochberg method.

**Figure 2 cancers-14-03931-f002:**
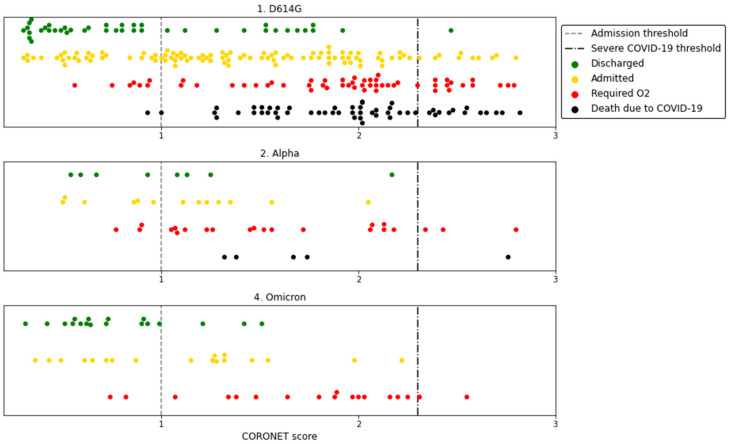
CORONET scores predicted for individual patients from wave 1. D614G, 2. Alpha, and wave 4. Omicron.

**Figure 3 cancers-14-03931-f003:**
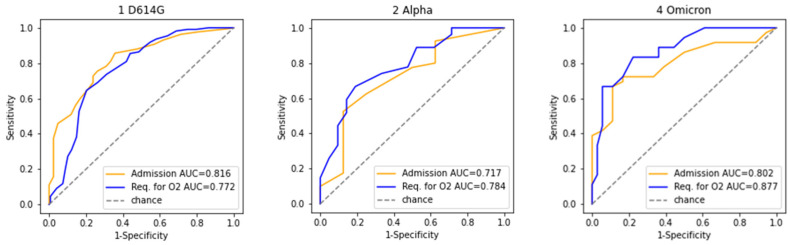
Performance of the CORONET model in different waves. CORONET scores were calculated for 258, 48, and 54 patients who were included from wave 1. D614G, 2. Alpha, and wave 4. Omicron. AUC = area under the curve calculated for admission and requirement for oxygen (O2).

**Table 1 cancers-14-03931-t001:** Patient characteristics during three SARS-CoV-2 waves.

		Unknown	Overall	1 D614G	2 Alpha	4 Omicron	
n			1968	1430	475	63	
Age *, median [Q1, Q3]		0	67.0 [57.0, 76.0]	68.0 [59.0, 77.0]	61.0 [50.0, 72.0]	66.0 [54.0, 73.0]	D614G vs. Alpha: *p* < 0.001, CLES ** = 0.621
Cancer type, n (%)	Breast	141	325 (17.8)	223 (17.3)	93 (19.6)	9 (14.3)	
	Colorectal		161 (8.8)	108 (8.4)	47 (9.9)	6 (9.5)	
	Lung		248 (13.6)	170 (13.2)	67 (14.1)	11 (17.5)	
	Other solid cancer		801 (43.9)	576 (44.7)	197 (41.6)	28 (44.4)	
	Haematological malignancy		291 (15.9)	212 (16.4)	70 (14.8)	9 (14.3)	
Solid cancer stage, n (%)	1	539	197 (17.3)	189 (24.5)	7 (2.2)	1 (1.9)	
	2		135 (11.9)	103 (13.4)	25 (7.9)	7 (13.5)	
	3		328 (28.8)	193 (25.1)	123 (38.9)	12 (23.1)	
	4		478 (42.0)	285 (37.0)	161 (50.9)	32 (61.5)	
Chemotherapy, n (%)		314	653 (33.2)	377 (26.4)	249 (52.4)	27 (42.9)	
Immunotherapy, n (%)		314	92 (4.7)	54 (3.8)	33 (6.9)	5 (7.9)	
Targeted Therapy, n (%)		314	197 (10.0)	121 (8.5)	64 (13.5)	12 (19.0)	
Radiotherapy, n (%)		486	106 (5.4)	50 (3.5)	45 (9.5)	11 (17.5)	
Vaccination, n (%)	Vaccinated 1 dose 2 doses 3 doses	15	-	0	0	46 (95.8) 1 (2.2) 8 (17.4) 37 (80.4)	
	Unvaccinated		-	1430 (100.0)	475 (100.0)	2 (4.2)	

* Statistical differences were calculated only for age because there were too many missing values for other categorical features; ** CLES = common language effect size, the probability that a score selected randomly from one distribution will be greater than a score selected randomly from another.

**Table 2 cancers-14-03931-t002:** Patient outcomes during three SARS-CoV-2 waves.

		Wave				
		Overall	1 D614G	2 Alpha	4 Omicron	Significant Differences between Waves *
n		1968	1430	475	63	
Outcome, n (%)	Discharged	548 (27.8)	336 (23.5)	186 (39.2)	26 (41.3)	1 vs. 2: *p* < 0.001; 1 vs. 4: *p* = 0.002
	Admitted	527 (26.8)	379 (26.5)	129 (27.2)	19 (30.2)	
	Admitted+O_2_	454 (23.1)	338 (23.6)	98 (20.6)	18 (28.6)	
	Admitted+O_2_+ died	439 (22.3)	377 (26.4)	62 (13.1)	0	1 vs. 2: *p* < 0.001

* Chi-squared tests; *p* values adjusted for multiple comparisons using the Benjamini–Hochberg method.

**Table 3 cancers-14-03931-t003:** Treatments for COVID-19.

	1 D614G			2 Alpha			4 Omicron		
	No. Patients Treated	No. Patients Not Treated	Missing	No. Patients Treated	No. Patients Not Treated	Missing	No. Patients Treated	No. Patients Not Treated	Missing
steroids	36 (32.4%)	75 (67.6%)	1319	88 (51.2%)	84 (48.8%)	303	19 (41.3%)	27 (58.7%)	17
remdesivir	1 (0.9%)	110 (99.1%)	1319	14 (8.3%)	155 (91.7%)	306	0 (0%)	45 (100%)	18
lipinovir	16 (14.4%)	95 (85.6%)	1319	0 (0%)	167 (100%)	308	0 (0%)	46 (100%)	17
interferon	0 (0%)	108 (100%)	1322	0 (0%)	167 (100%)	308	0 (0%)	46 (100%)	17
interferon beta	4 (3.6%)	106 (96.4%)	1320	0 (0%)	167 (100%)	308	0 (0%)	46 (100%)	17
anticoagulation prophylaxis	70 (64.2%)	39 (35.8%)	1321	79 (46.7%)	90 (53.3%)	306	22 (47.8%)	24 (52.2%)	17
anticoagulation treatment dose	6 (5.5%)	103 (94.5%)	1321	20 (11.9%)	148 (88.1%)	307	0 (0%)	46 (100%)	17
antibiotic	79 (71.8%)	31 (28.2%)	1320	100 (58.8%)	70 (41.2%)	305	17 (37%)	29 (63%)	17
plasma	2 (1.8%)	108 (98.2%)	1320	5 (3%)	162 (97%)	308	0 (0%)	46 (100%)	17
tocilizumab	5 (4.5%)	105 (95.5%)	1320	8 (4.8%)	159 (95.2%)	308	2 (4.3%)	44 (95.7%)	17
nebulised interferonb	1 (0.9%)	106 (99.1%)	1323	0 (0%)	167 (100%)	308	0 (0%)	46 (100%)	17
hydroxychloroquine	51 (45.9%)	60 (54.1%)	1319	0 (0%)	166 (100%)	309	0 (0%)	46 (100%)	17
aspirin	1 (1%)	102 (99%)	1327	2 (1.2%)	165 (98.8%)	308	0 (0%)	46 (100%)	17
baricitinib	0 (0%)	15 (100%)	1415	0 (0%)	49 (100%)	426	0 (0%)	46 (100%)	17
molnupiravir	-	-	1430	-	-	475	0 (0%)	42 (100%)	21
sotrovimab	-	-	1430	-	-	475	0 (0%)	42 (100%)	21
other drug	-	-	1430	-	-	-	-	-	63

**Table 4 cancers-14-03931-t004:** Comparison of patients’ characteristics associated with severity of COVID-19 during waves, stratified by outcome.

		Wave, Median [Q1, Q3]			
Variable	Outcome	1 D614G	2 Alpha	4 Omicron	Significant Differences between Waves *
Age	Discharged	62.0 [53.0, 71.0]	58.0 [44.0, 67.8]	66.0 [60.5, 72.0]	1 vs. 2 *p* = 0.0017; 2 vs. 4 *p* = 0.0473
	Admitted	65.0 [56.0, 74.0]	61.0 [51.0, 71.0]	53.0 [43.5, 73.0]	1 vs. 2 *p* = 0.0273
	Admitted+O_2_	71.0 [63.0, 78.0]	68.0 [55.0, 76.0]	67.5 [64.2, 74.5]	1 vs. 2 *p* = 0.0125
	Admitted+O_2_+died	73.0 [65.0, 81.0]	70.5 [58.2, 78.0]		1 vs. 2 *p* = 0.0034
Solid cancer stage	Discharged	2.5 [1.0, 3.0]	3.0 [3.0, 4.0]	4.0 [3.0, 4.0]	1 vs. 2 *p* = 0.0000, 1 vs. 4 *p* = 0.0041
	Admitted	3.0 [2.0, 4.0]	4.0 [3.0, 4.0]	4.0 [3.0, 4.0]	1 vs. 2 *p* = 0.0001
	Admitted+O_2_	3.0 [1.0, 4.0]	4.0 [3.0, 4.0]	4.0 [3.0, 4.0]	1 vs. 2 *p* = 0.0000, 1 vs. 4 *p* = 0.0146
	Admitted+O_2_+died	4.0 [2.0, 4.0]	4.0 [3.0, 4.0]		1 vs. 2 *p* = 0.0125
Performance status	Discharged	1.0 [0.0, 1.0]	1.0 [1.0, 1.0]	1.0 [0.2, 1.0]	
	Admitted	1.0 [1.0, 2.0]	1.0 [1.0, 2.0]	1.0 [0.0, 1.0]	
	Admitted+O_2_	1.0 [1.0, 2.0]	1.0 [1.0, 2.0]	2.0 [1.0, 2.0]	
	Admitted+O_2_+died	2.0 [1.0, 3.0]	1.0 [1.0, 2.0]		1 vs. 2 *p* = 0.0075
Total no. comorbidities	Discharged	1.0 [1.0, 2.0]	1.0 [1.0, 2.0]	1.0 [1.0, 2.0]	
	Admitted	1.0 [1.0, 3.0]	1.0 [0.0, 2.0]	1.0 [0.0, 2.0]	
	Admitted+O_2_	2.0 [1.0, 3.0]	1.0 [1.0, 2.0]	1.0 [1.0, 2.8]	1 vs. 2 *p* = 0.0014
	Admitted+O_2_+died	2.0 [1.0, 4.0]	2.0 [1.0, 3.0]		
NEWS2	Discharged	1.0 [0.0, 2.0]	1.0 [0.0, 3.0]	1.0 [0.0, 1.0]	
	Admitted	2.0 [1.0, 3.0]	2.0 [1.0, 4.0]	1.0 [0.0, 2.0]	
	Admitted+O_2_	4.0 [2.0, 6.0]	3.0 [1.0, 4.0]	3.5 [2.0, 5.0]	1 vs. 2 *p* = 0.0063
	Admitted+O_2_+died	4.0 [2.0, 7.0]	5.5 [3.0, 8.0]		
Albumin	Discharged	40.0 [36.0, 43.0]	41.0 [36.0, 44.0]	40.0 [36.0, 43.2]	
	Admitted	37.0 [32.0, 41.0]	39.0 [34.0, 42.0]	37.0 [31.5, 43.0]	
	Admitted+O_2_	33.0 [28.0, 37.0]	30.0 [23.5, 35.5]	30.5 [25.0, 36.0]	1 vs. 2 *p* = 0.0029
	Admitted+O_2_+died	32.0 [26.0, 38.0]	31.0 [28.0, 38.0]		
C-reactive protein	Discharged	12.0 [4.6, 47.0]	9.5 [3.0, 39.8]	4.3 [3.1, 17.0]	
	Admitted	36.0 [12.0, 90.4]	22.0 [6.4, 65.2]	43.6 [7.8, 108.9]	
	Admitted+O_2_	70.0 [36.3, 127.0]	57.1 [18.5, 122.4]	62.4 [25.7, 128.1]	
	Admitted+O_2_+died	91.0 [40.0, 168.0]	118.8 [52.8, 201.7]		
Lymphocytes	Discharged	1.1 [0.7, 1.9]	1.3 [0.9, 1.8]	1.1 [0.6, 1.6]	
	Admitted	0.9 [0.5, 1.4]	0.9 [0.6, 1.6]	0.8 [0.5, 1.6]	
	Admitted+O_2_	0.9 [0.6, 1.4]	0.8 [0.6, 1.2]	1.1 [0.6, 1.4]	
	Admitted+O_2_+died	0.7 [0.4, 1.2]	0.6 [0.4, 1.2]		
NLR	Discharged	2.9 [1.9, 5.1]	2.4 [1.6, 4.2]	3.3 [1.7, 5.9]	
	Admitted	3.7 [1.9, 8.6]	3.9 [2.1, 7.5]	4.3 [2.6, 11.1]	
	Admitted+O_2_	4.5 [2.7, 9.3]	4.6 [2.1, 7.6]	5.4 [2.9, 6.8]	
	Admitted+O_2_+died	5.9 [3.1, 13.0]	6.6 [3.1, 12.3]		
Neutrophils	Discharged	3.5 [2.2, 4.9]	3.2 [2.2, 4.5]	3.5 [2.0, 4.2]	
	Admitted	3.9 [2.2, 6.3]	3.7 [2.2, 5.8]	3.9 [3.0, 9.2]	
	Admitted+O_2_	4.4 [2.9, 6.8]	4.0 [2.1, 5.4]	5.3 [3.6, 8.3]	
	Admitted+O_2_+died	5.0 [3.0, 7.9]	5.3 [2.6, 9.6]		
Platelets	Discharged	217.0 [170.0, 271.0]	229.0 [188.0, 275.0]	193.0 [130.5, 285.5]	
	Admitted	216.0 [149.2, 296.0]	234.0 [151.8, 311.5]	243.0 [196.5, 389.0]	
	Admitted+O_2_	192.0 [143.0, 264.0]	204.0 [143.0, 275.2]	202.5 [157.5, 312.8]	
	Admitted+O_2_+died	183.5 [110.0, 270.0]	181.0 [127.0, 287.0]		

* Mann–Whitney U and Chi2 tests were used for numeric and categorical (solid cancer stage and performance status) features, accordingly; *p*-values were corrected for multiple testing, using one-step Sidak correction.

## Data Availability

Within the constraints of the data agreements between the University of Manchester and other institutions, data are available upon request.

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
