# Peer review of "An International Comparison of Presentation, Outcomes and CORONET Predictive Score Performance in Patients with Cancer Presenting with COVID-19 across Different Pandemic Waves"

_cancers, 2022, doi:10.3390/cancers14163931_

Round 1

Reviewer 1 Report

Well done and interesting paper to read and to know about 

Author Response

Well done and interesting paper to read and to know about 

Response: We thank the reviewer for their positive comments.

Reviewer 2 Report

Dear Authors,

I read your work entitled “Analysis of differences in presentation and outcome of patients with cancer presenting with COVID-19 during different waves of the pandemic” and here I enclose my recommendations to you:

A) Introduction

The introduction is too short. I suggest you update it so the rational of this work to be sounder to the readers of this work.

B) Methods

Please, make clear to the reader how the sample was recruited and included in this study (e.g. number of countries, number of hospitals etch.) that would make the readers understand better your Methodological plan. You make reference to the title of this work about outcomes and here some weaknesses that arise from that (a) what was the methods to lead to your outcomes (e.g., statistical plan), (b) what was the criterion/s of the outcomes? Generally, when we measure outcomes in health care the main measures are: Effectiveness of care, Efficient use of medical imaging or blood tests, Mortality, Patient experience, Readmissions, Safety of care, Timeliness of care and many more.

Thank you.

Author Response

Dear Authors,

I read your work entitled “Analysis of differences in presentation and outcome of patients with cancer presenting with COVID-19 during different waves of the pandemic” and here I enclose my recommendations to you:

  1. A) Introduction

The introduction is too short. I suggest you update it so the rational of this work to be sounder to the readers of this work.

Response: Thank you - we have amended the introduction and included further background regarding vaccination and rationale regarding the study.

  1. B) Methods

Please, make clear to the reader how the sample was recruited and included in this study (e.g. number of countries, number of hospitals etch.) that would make the readers understand better your Methodological plan. You make reference to the title of this work about outcomes and here some weaknesses that arise from that (a) what was the methods to lead to your outcomes (e.g., statistical plan), (b) what was the criterion/s of the outcomes? Generally, when we measure outcomes in health care the main measures are: Effectiveness of care, Efficient use of medical imaging or blood tests, Mortality, Patient experience, Readmissions, Safety of care, Timeliness of care and many more.

Thank you.

Response:

A large amount of detail regarding the data collection and outcomes is available in our previously published paper, which we reference in this paper. However, to provide more details about the patients recruited to the study we have added a paragraph in the ‘Methods’:

“Patient data was collected worldwide from the United Kingdom, the United States, Spain, Denmark, and France and collectively from medical centers contributing to ESMO-CoCARE. We aimed to include a range of hospitals managing cancer patients in different settings from local district general/local community hospitals in which general acute physicians manage acute oncology admissions, to tertiary cancer centres where there are highly specialised oncology services.

” and provided a Supplementary Table 8 containing detailed no. patients in each cohort in a given wave. We added this information in the first paragraph in ‘Results’ section.

To clarify the definition of outcomes used in our analysis we added a paragraph (in ‘Methods’):

“We previously developed a tool (CORONET) to help determine the need to admit a patient to hospital on the basis of their likelihood of needing O2 (as generally it is only given in hospital) and the severity of COVID-19 indicated by prediction for O2 re-quirement and death (17). As a result, we obtain 4 key outcomes, arranged in a 0-3 point ordinal scale: discharged, admitted (≥ 24 hours inpatient), admitted+O2 require-ment (including ventilator support), and admitted+O2+died (directly attributable to COVID-19 disease, not cancer), which were used as measures of disease severity (17). Contrary to an analysis of binary outcomes (e.g. O2 vs no need for O2), this strategy im-proves generality to reflect the whole clinical picture; important for overall decision-making regarding hospital admission. Thresholds of >=1.0 determine admission and >=2.3 determine severe outcome (more details are available in (17)).  ”

Reviewer 3 Report

REVIEW REPORT

Title: An international comparison of presentation, outcomes and CORONET predictive score performance in patients with cancer presenting with COVID-19 across different pandemic waves

NO: cancers-1790646

Journal: Cancers, MDPI

Section: Cancer Epidemiology and Prevention

Date: 04.07.2022

Conflict of interest statement for peer reviewers:

I declare that I do not have any competing interests

Structure and content:

-          Abstract – please see the recommendations for authors, the abstract is too long (200 words max – yours is over 350 words long)

-          Why is there a blank page with a lonesome “1.” after the front page?

-          Where is the “side matter” (article no., name of article, and authors? – please look at the template)

-          … the majority was PCR-based – what does majority mean, and how many exactly? – such a phrase is not contributing to the exactness of the article at hand

-          I have not seen it mentioned but are all these cases unique or are there patients who were infected more than 1 time?

-          Did the authors experience any changes in tumor growth in the patient cohort (e.g., unusual tumor regression following a vaccination, https://jitc.bmj.com/content/10/3/e004371)?

-          Since the authors partly used data from a previous article, could they potentially enrich the manuscript with 30-day mortality data or some sort of follow-up?

Language and formatting-related comments:

-          Please check the spelling and formatting of your author list (e.g., the affiliations are always being numbered and given in ascending order, one does not start with the first author and state affiliations 2 and 3)

-          Where are the line numbers?

-          Furthermore, after the author Timothy Cooksley there is ESMO Co-Care written, any particular reason for this? If there is an additional list of authors, then I have not seen it anywhere

-          The name André has an acute accent (according to the submission system) – there is none in the article – please check the names of your co-authors thoroughly. This is common decency.

-          Check punctuation – in the affiliations list there are sometimes full stops and sometimes none, also sometimes the land of origin is omitted, in most instances, it is stated, please stay consistent

-          Please either use COVID-19 infection or SARS-CoV-2 infection, the authors should stay consistent throughout the whole text

-          Most major style guides (including APA, MLA, the Chicago Manual of Style, and Harvard) do not require et al. to be italicized

-          The waves are not named consistently – perhaps either begin with wave no. wave 1 XXX, wave 2 XXXX, wave 3 XXX, or start with the name of the variant and put the wavenumber in brackets

-          “Until present” regarding the duration of the last wave? One should exactly state the cut-off date. I presume the authors stopped collecting data at some point to analyze and prepare the article

-          How many patients were there during phase 3? – was data not sufficiently collected?

-          Table 1 has no title

-          Table 1 – the therapy regime is not sufficiently clear. Are the immunotherapy patients included in the chemotherapy patients? For chemotherapy, immunotherapy, and targeted therapy there are always 314 cases missing, for radiotherapy suddenly 486?

-          The readership would be probably interested in how many patients were treated and how many were not treated (dichotomous). Then, looking at the treated cases - how many were primary surgery / surgery with neoadjuvant therapy / only chemotherapy + radiotherapy? A graphic might help.

-          Also, what stage is presented with “solid cancer stage”? Is it the pT stage or the cT stage?

-          I presume the tumors were of different histological types and from different organs, right? A summary should be added

-          Furthermore, if that is the case, I do not believe all tumors should be lumped together. This also raises concerns regarding the design of the study

-          Vaccination - at least one dose? For the sake of clarity, I would use different terminology – unvaccinated, partially, and fully vaccinated. The corresponding numbers of patients (n) would be helpful.

-          It would be helpful to know what percentage of all COVID-19 positive cases were patients with cancer (percentage)

-          Table 3 – somewhat messy – looking at the first line (and some others as well) – how come the first wave has according to the steroid no. of treated patients suddenly 1431 cases in total? The numbers do not add up

-          Table 3- how come the molnupiravir, sotrovinab and other drug lines are sometimes empty and sometimes not?

-          PPatients – typo

-          Different versions of spelling O2 vs O2 – lack of consistency

-          COVID-19severity – lack of spacing

-          Lack of acknowledgments, conflict of interest statement, funding, etc.

-          The font of the references appears bigger than the rest, please check

-          The literature does not adhere to the standard citation style – please check other publications or download the citation style and use a citation manager (turn off the option to add URL when citing journals / cite URL only when webpages are being used)

Presentation:

-          The readership might welcome a timeline of collection and variants in the Materials and methods section

-          The supplementary tables are somewhat crowded and hard to read – perhaps use a different orientation (landscape)

-          Please check the template of the journal (see comments above)

Opinion:

The authors present an additional validation study of their scoring system and a comparison between waves.

Some concerns:

-          The distribution of patients regarding their underlying disease (histology, type) and their need for further treatment. A patient with pancreatic cancer has surely a different prognosis than a patient with ER+/PR+ breast cancer or someone with CRC.

-          The admission protocol and treatment regime. It seems multiple institutions were included. Did all adhere to the same protocol at admission and treatment regime? If not, what differed? Did those cohorts exhibit any differences? How was this multi-centric study supervised?

-          How did the treatment regime, in general, differ between the individual waves?

-          Were any changes in the primary disease observed?

-          How about comorbidities? One would assume that concurrent comorbidities are independently associated with severe COVID-19 or death. I know that the number of comorbidities is already included in the CORONET score. Nevertheless, it would be interesting to know specifically which ones.

I believe the article should undergo at least a major revision (or rejection with resubmission). The authors should incorporate an additional novel aspect, survival statistics, a comparison of the patients' outcome in regard to their underlying disease, or perhaps something different.

With best regards.

Round 2

Reviewer 2 Report

Dear Authors, 

I read your work and I can say that several improvements were addressed. I will comment again that the "Introduction" does not include a sound rational even after the new adds and I suggest you to update it.

Thank you.

Reviewer 3 Report

REVIEW REPORT

Title: An international comparison of presentation, outcomes and CORONET predictive score performance in patients with cancer presenting with COVID-19 across different pandemic waves

NO: cancers-1790646

Journal: Cancers, MDPI

Section: Cancer Epidemiology and Prevention

Date: 06.08.2022

Conflict of interest statement for peer reviewer:

I declare that I do not have any competing interests

Report

The structure has been improved.

The questions have been answered. Sadly, not all with factual data. However, I understand there are limitations to the study design. I would include some of my previous inquiries in the limitation list of the article.

Certain typos persist (spacing etc.).

Percentages in the vaccination row are not entirely clear to me (Table 1).

The results and presentation are satisfactory.

I suggest a minor (language) revision.

With best regards.